# A Positive-Reinforcement Training Regimen for Refined Sample Collection in Laboratory Pigs

**DOI:** 10.3390/ani15040471

**Published:** 2025-02-07

**Authors:** Rachel Layton, David Beggs, Andrew Fisher, Peter Mansell, Sarah Riddell, Daniel Layton, David T. Williams, Kelly J. Stanger

**Affiliations:** 1CSIRO, Australian Centre for Disease Preparedness, Geelong, VIC 3219, Australia; sarah.riddell@csiro.au (S.R.); d.williams@csiro.au (D.T.W.); kelly.stanger@csiro.au (K.J.S.); 2Melbourne Veterinary School, Faculty of Science, University of Melbourne, Werribee, VIC 3030, Australia; dbeggs@unimelb.edu.au (D.B.); pmansell@unimelb.edu.au (P.M.); 3CSIRO, Health and Biosecurity, Geelong, VIC 3219, Australia; daniel.layton@csiro.au

**Keywords:** swine, welfare, anaesthesia, restraint, stress, research

## Abstract

Pigs are a commonly utilised animal in laboratory research. A requirement for many pig studies is the collection of biological samples, such as oral and rectal swabs. This commonly requires pigs to be physically restrained or anaesthetised to ensure staff safety and successful sample collection. Pigs can be trained to voluntarily engage in sample collection by being restrained in a sling, but this training requires a time investment that is not practical for many studies. We developed a training regimen using positive reinforcement that can be used for both group-housed and individually housed pigs to collect samples from pigs voluntarily and unrestrained. Pigs were trained within six days (group housed) and 18 days (individually housed) for the collection of swabs and rectal temperatures, with a time investment of approximately 5 min/pig/day. The pigs undergoing daily training displayed behaviours and salivary corticosterone levels indicating positive animal welfare and low levels of stress. This training regimen provides a practical option for increasing the use of positive-reinforcement training in laboratory pigs to reduce the need for forced restraint and improve the welfare of pigs in research.

## 1. Introduction

Despite the developments in in vitro and ex vivo technologies, the complexities of physiology and immunity mean that in vivo animal models remain essential [1]. Sound laboratory animal welfare is necessary for producing valid and repeatable scientific results [2] and for maintaining the societal licence afforded to research institutions to use animals for scientific purposes [3]. Therefore, researchers have an ethical obligation to adhere to and continually refine methods that optimise research animal welfare.

Pigs are a common model for human health and animal disease research due to their similar anatomical structure to humans and importance in food production [4]. The size and strength of pigs mean that advancing best-practice methodology for laboratory pig welfare and management has unique difficulties that are not as commonly encountered with smaller, more traditional laboratory species [5]. Forcible manual restraint such as snaring is routinely utilised for sample collection. Whilst snaring can serve as a rapid and reliable method for single or occasional restraint events, aversion can quickly develop in pigs snared repeatedly [6]. In these instances, an alternative to stressful forced restraint is to anaesthetise pigs for sample collection. Whilst this has the potential to enhance welfare by reducing stressful restraint, the induction of anaesthesia can alter physiology and immunological parameters and, in turn, study outcomes [7].

Training laboratory pigs (including mini-pigs) can reduce the reliance on chemical and forced manual restraints. Yang et al. 2021 trained pigs to be lifted into a pig sling restraint: a metal frame containing a hammock. The pigs were trained for 30–60 min per day to allow for experimentally induced wound care and blood collection without the need for forcible or chemical restraint [8]. Whilst this time investment in daily training is arguably justified for studies of this nature, for shorter-term pig studies without complex wound care or sample requirements, this investment of time is not practical. The perceived and potentially significant time investment required to train pigs is a potential reason for the limited implementation to date. Additional barriers to the adoption of positive-reinforcement training regimens in laboratory pigs are the need to lift pigs into a sling restraint [8,9], which is not practical for larger pigs, in addition to the cost of specialised sling equipment.

To address these issues, we developed a rapid, three-phase, positive-reinforcement training regimen for use in both individually housed and group-housed laboratory pigs to allow for minimally invasive samples to be collected from conscious pigs voluntarily. The ability to train pigs to have oral swabbing, rectal swabbing and rectal thermometer insertion was assessed by documenting the time to complete each phase of training in both individually housed and group-housed pigs with and without the need to restrain pigs in a pig sling. Additionally, the behaviour of individually housed pigs was assessed by conducting a daily human approach test, and salivary corticosterone was analysed.

## 2. Materials and Methods

### 2.1. Study Design

Two independent studies were conducted to determine if a novel positive-reinforcement training (PRT) regimen would facilitate voluntary conscious sample collection in both individually housed (study 1) and group-housed (study 2) pigs. Both studies were performed in 6-week-old Landrace cross female pigs sourced from the same local commercial piggery (Geelong, Victoria). Individually housed pigs (n = 4) were maintained in a pen measuring 1.9 m × 2.96 m within sight, sound and smell of one another. An additional untrained pig was included as an observational comparison under the same housing conditions. Group-housed pigs (n = 3) were maintained in a pen measuring 7 m × 2.96 m, with a total of six pigs present in the pen. Training success was measured by documenting the time taken for each pig to complete daily PRT (phase 1, 2 and 3). In addition to PRT, saliva collection for corticosterone analysis and a human approach test (HAT) were performed daily on individually housed pigs. Group-housed pigs were part of a separate larger study; therefore, these variables were not measured.

### 2.2. Animal Housing, Husbandry and Daily Routine

Each pig floor pen was furnished with enrichment items (plastic balls, chains and rubber hosing) that were rotated daily, a plastic bed containing straw and a rubber mat. Barastoc™ pig grower pellets (Ridley Corporation, Melbourne, Australia) were available ad libitum. Room temperature was maintained at 22 °C, and lights were maintained on an 8 h light/16 h dark cycle. Prior to the initiation of PRT, the same operator conducted husbandry, HAT and saliva collection from the day after arrival, with a second operator assisting. For individually housed pigs, all steps were conducted in full before moving to the next pen. Husbandry commenced each day at 9 am and took approximately 10 min per pen. Upon room entry, two operators verbally greeted the pigs, removed all furnishings and hosed the pen thoroughly. A 3 min HAT was then conducted, followed by saliva collection (1–3 min) and 5 min spent calmly interacting with the pig before 2–5 min of PRT commenced. For the group-housed pigs, husbandry was conducted (approximately 20 min) followed by 5 min of calm interaction before commencing 2–5 min of PRT per pig.

### 2.3. Human Approach Test for Behavioural Assessment

After husbandry, video recording commenced from outside of the pen. The operator (an experienced animal technician) entered the pen and conducted a three-minute human approach test, as described by Hemsworth et al. 1981, by standing silently in the pen corner [10]. The operator moved their legs up and down slowly and rhythmically to prevent bite injury during the three-minute HAT. A behavioural ethogram was developed using the principles described by Claxton et al. 2011 to assess the behaviours displayed by the pigs during the HAT [11]. The ethogram was adapted and modified for the present study to ensure all behaviours displayed during the HAT were captured and categorised as either positive or negative. Each behaviour was assigned a score of 1, 2 or 3 based on the weighting described in Table 1, with total daily scores per pig then calculated for both negative and positive behaviours. A behaviour that lasted longer than 5 s was counted as a new behaviour to capture its greater significance (e.g., longer grunts and squeals). All HAT footage was retrospectively reviewed, and the behavioural ethogram was completed by the same operator. The operator was not blinded, and the footage review was not randomised. A second operator was utilised to randomly select HAT footage from one pig per day and conduct the behavioural ethogram to confirm observer reliability. All assessments were confirmed to be consistent between operators.

### 2.4. Saliva Collection for Corticosterone Analysis

Saliva was collected daily by allowing the pigs to voluntarily chew on Sarstedt Salivette™ swabs for cortisol (Numbrecht, Germany) clamped in haemostats for 1–3 min (until visibly saturated). All saliva collections occurred simultaneously for each pig each day, and the saliva samples were stored at 4 °C until processed. The samples were centrifuged (4 °C, 2 min, 1000× *g*) within 12 h of collection and stored at −80 °C. The samples were then thawed to room temperature before analysis and then analysed using 96-well corticosterone ELISA kits according to the manufacturer’s instruction (catalogue No. ADI-900-907) by ENZO Life Sciences (Farmingdale, NY, USA).

### 2.5. Positive-Reinforcement Training (PRT)

The pigs did not have any prior exposure to positive-reinforcement training and could see each other during the daily training sessions. The habituation of the pigs to the operator, use of a paddle for target training and training pigs to enter the Panepinto sling were based on the work of O’Malley et al. 2022 [9]. This was then modified to a three-phase regimen to facilitate voluntary Panepinto sling entry and sample collection, as opposed to lifting pigs into a sling for restraint. From phase 1, each time a desired behaviour was performed, the pig was positively rewarded with a verbal cue (Yay!) and food reward (pumpkin cut into 1 cm × 5 cm pieces or strawberry yoghurt administered via a 20 mL syringe). In phase 1, the pigs were trained to touch a wooden paddle with their snout; in phase 2, the pigs were trained to enter a Panepinto sling (Colorado, USA) (Figure 1a); and in phase 3, the pigs were trained to remain stationary for the collection of oral swabs, rectal swabs and rectal temperatures, either raised in the Panepinto sling (Figure 1b) or standing for the duration of sample collection (Figure 1c).

PRT was conducted within the home pens of the individually housed pigs. The group-housed pigs were separated into a pen of the same dimensions as an individual pig pen attached to the group pen and maintained visual and auditory contact with the group during PRT. For each phase, the time to complete training was categorised as fast (0–30 s, ‘rapid approach’), medium (31–60 s), slow (61–180 s) or unsuccessful (>180 s). If a phase of training was not completed within 180 s, PRT was deemed unsuccessful and ceased for the day.

**Training habituation:** The operator spent 5 min sitting in the pen and talking to the pigs to encourage voluntary approach. The pigs would receive pats and a food reward upon approaching the operator. This 5 min period was completed in full even if the pigs did not approach or engage with the operator and was discontinued after successful completion of phase 1.

**Phase 1: Touching a paddle to receive a reward.** A wooden paddle was held against the bars of the pen by the operator who remained outside of the pen. The verbal cue and a reward were given each time the pig touched the paddle with its snout. If a pig touched the paddle within the 180 s training period, 5 additional touches and rewards were conducted to reinforce the desired behaviour and training ceased. Successful completion of phase 1 was defined as a pig rapidly touching the paddle (within 30 s of commencing PRT). Progression to phase 2 occurred the day after successful completion of phase 1.

**Phase 2: Entry into the Panepinto sling.** Phase 2 commenced with completion of phase 1, followed by opening the pen gate to expose the entrance to the wound-down Panepinto sling. A paddle was used as a lure to encourage the pig to enter the Panepinto sling by stepping over a low horizontal bar (7 cm) at the entrance to the sling. The paddle was advanced in 30 cm increments until the pig was standing fully within the Panepinto sling. If a pig did not voluntarily move forward to touch the paddle, the paddle was raised for 2 s and then placed back down approximately 15 cm closer to the pig to encourage progress. A reward was provided once if a pig touched the paddle without stepping forward, after which the pig was only allowed to touch the paddle for a reward if it stepped forward. Once the pig had stepped all four feet into the Panepinto sling, the paddle was placed directly in front of the pig to allow for a stationary paddle touch, at which time the paddle was lifted up and away whilst the reward was provided. Once the reward had been consumed, the paddle was placed back in front of the pig for additional paddle touches and rewards. This process was repeated for 10 s, after which training ended, and the pig was encouraged out of the Panepinto sling back into the pen using the paddle as a lure. The paddle was then presented a final time for the pig to touch and receive a reward. Successful completion of phase 2 was defined as the pig walking fully into the Panepinto sling and remaining for 10 s. Progression to phase 3 occurred the day after successful completion of phase 2.

**Phase 3: Remaining stationary for sample collection.** Phase 3 included completion of phase 1 and phase 2, with the paddle being used as a guiding tool to position the pig over the leg holes in the sling. If the pigs moved too far back, the pig was required to move forward and touch the paddle to receive a reward. If the pigs moved too far forward in the Panepinto sling, the operator used the palm of their hand to place gentle pressure on the pigs’ forehead whilst saying ‘backup’. Once in the correct position, the paddle was presented for the pig to touch and receive a reward. The paddle was then placed directly in front of the pig to touch, lifted whilst a reward was provided and placed back in front of the pig when it was finishing the previous treat, whilst a second operator wound the sling up. If a pig did not struggle, the paddle continued to be presented to the pig for rewards, whilst swabs and rectal temperatures were taken whilst raised in the sling. If a pig struggled, the sling was wound down while the operator continued to present the paddle for rewards, and swabs and rectal temperatures were collected from the stationary standing pig. If the pig moved away from the paddle, the operator maintained the paddle position for the pig to return to, after which the paddle touching and rewards recommenced until the samples had been collected. Successful completion of phase 3 was defined as pigs completing phase 1 and 2 and standing still or being restrained in the Panepinto sling for sample collection.

## 3. Results

Using the three-phase positive-reinforcement training (PRT) regimen described, oral and rectal swabs and rectal temperatures were collected from individually housed pigs (n = 3) within 18 days and within six days for group-housed pigs (n = 3) from initiation of training (Figure 2). Overall, there was more rapid and consistent training success in the group-housed pigs compared with the individually housed pigs. Three out of four individually housed pigs were successfully phase 1 trained to ‘rapid approach’ by day three, and all four pigs were phase 1 trained to rapid approach by day five, while all three group-housed pigs were phase 1 trained to rapid approach from day three (Figure 2a). For phase 2 of PRT, three out of four individually housed pigs rapidly entered the Panepinto sling by day seven onwards; however, one pig could not be successfully trained to enter the Panepinto sling at any timepoint and, therefore, did not progress to phase 3 of PRT (Figure 2b). In comparison, all three group-housed pigs achieved rapid Panepinto sling entry (phase 3) from day four onwards (Figure 2b). Of the three individually housed pigs that progressed to phase 3 of PRT, successful training to allow for conscious sample collection occurred by day 18 (Figure 2c) compared to the group-housed pigs, which were trained by day six (Figure 2c). While the pigs had variable responses to restraint in the Panepinto sling even if they were individually or group housed, all six pigs that completed phase 3 of training were consistently and successfully sampled while standing unrestrained. Collecting samples from free-standing pigs was more successful than raising pigs in the sling, with only one pig (individually housed) raised daily without struggling. After two failed attempts of raising a pig in the sling, the samples were collected free-standing for the remainder of the study.

For the individually housed pigs, a human approach test (HAT) was conducted daily, and behaviours were assessed via an ethogram, with trained pigs demonstrating increasing positive behaviours and no or very few negative behaviours towards the operator (Figure 3). Behaviours during a HAT were also observed for a single untrained pig. Although this can be observational only, the single untrained pig was observed to display high and fluctuating negative behaviour scores from study day three onward (Figure 3a). Whilst the untrained pig displayed a lower positive behaviour score compared to the mean positive behaviour score of the trained pigs, the high individual variation between the trained pigs makes the comparison between the trained pigs and the untrained pig unreliable (Figure 3b). In addition to the behavioural observations during the daily HAT via an ethogram, daily salivary corticosterone concentrations of all pigs throughout the study were assessed. All levels remained below 1500 pg/mL except for day 1, where one pig displayed levels above 4000 pg/mL (Figure 3c).

## 4. Discussion

This study assessed a rapid, positive-reinforcement training (PRT) regimen to facilitate voluntary, conscious sample collection in both individually housed and group-housed pigs without the need to lift pigs into a sling restraint or a heavy time investment in training. By using this three-phase PRT regimen, oral and rectal swabs and rectal temperatures were collected in conscious laboratory pigs without the requirement for manual or chemical restraint. More rapid and consistent training success occurred in the group-housed pigs (six days) compared to the individually housed pigs (18 days).

The results reported in this study reflect the current literature. Pigs are known to respond well to positive-reinforcement training, as demonstrated in a study by Brajon et al. 2015 [12]. The authors treated pigs with either positive, negative or neutral human interactions, with response to humans scored using human approach and reactivity tests. The authors observed that only the pigs undergoing positive human interactions of food rewards and gentle handling approached and consistently touched the operator. While these positive responses of pigs to PRT techniques are established in the literature, the present study is the first to demonstrate and quantify the differences in training responses between individually housed and group-housed pigs. One explanation for the faster and more consistent training success observed in the group-housed pigs may be increased competition for food, which has been shown to be a key driver of behaviour in pigs. This was described by Boumans et al. 2018, who found that, as pig group sizes increased, so did the rate of feed consumption, whilst the time spent feeding decreased [13]. This indicates an increased feeding urgency in the presence of other pigs, which, for the present study, possibly explains the faster training progression observed for the group-housed pigs.

The ability to collect minimally invasive samples from the group-housed pigs within six days using this regimen negates the need to extend typical pre-study acclimation periods for incorporating PRT. As extending study timelines results in additional cost and time, the rapid training success of the pigs in the present study removes a current barrier to implementation. Future research may be useful to investigate if longer daily training sessions or multiple training sessions each day further reduce the time to training success. An additional potential barrier to the adoption of PRT in laboratory pigs is the requirement to purchase specialised restraint equipment. While the present study used a Panepinto sling to restrain the pigs for sample collection, all trained pigs could be reliably trained to stand voluntarily at ground level for the collection of swabs and rectal temperatures. Previous studies that have used PRT for restraint of laboratory pigs commonly utilised mini-pigs or smaller pigs (<45 kg) that can be lifted into a sling. Pigs have either demonstrated aversion to remaining restrained in the sling despite PRT over a 13-day period [9] or have required a greater time investment (30–60 min per day) to facilitate relaxed sling restraint [8]. This additional time and equipment investment is arguably warranted for longer-term studies or in studies involving more invasive sample collection. Whilst blood collection was not assessed in the present study, a likely limitation of this PRT regimen is that it would not allow for conscious blood samples to be collected from pigs. Further research into potential techniques to facilitate blood collection using this PRT regimen is recommended, such as direct collection from the ear vein for smaller blood volumes and the surgical implantation of vascular access ports for larger blood volumes. Yet, for minimally invasive sample collection, the present study suggests that this PRT regimen can negate the need to purchase sling equipment or invest additional time training pigs for sling restraint. This increases the accessibility and ease of incorporating PRT for laboratory pigs and provides an option for studies of larger pigs that cannot be easily lifted into a sling. By using this training regimen in free-standing pigs, samples can be collected from pigs that choose to voluntarily engage in the sample collection process, demonstrating the welfare benefits of this training regime.

In addition to the practical benefits of this PRT regimen, the pigs undergoing PRT in the present study displayed increasing positive behaviours, very few negative behaviours and salivary corticosterone levels within normal limits for non-stressed pigs [14]. The increasing positive behaviours of the trained pigs in the present study follow the published literature, as demonstrated in a study by Jonholt et al. 2021. The authors trained pigs using either paddle lure training or clicker training using an auditory cue. The pigs were then presented with a novel task participation test, and their behaviour was recorded. All trained pigs voluntarily participated in the test, with only one pig in the lure trained group showing reduced welfare indicators [15]. These findings reflect the increasing number of positive behaviours exhibited by the trained pigs in the present study and further demonstrate the apparent welfare benefits of incorporating PRT for laboratory pigs. The low number of negative behaviours, including bites to the operator, displayed by the pigs undergoing PRT in the present study also demonstrates safety benefits by greatly reducing the risk for operator injury. Whilst conclusions cannot be drawn due to there being only one untrained pig, the high number of negative behaviours displayed by this pig starkly contrasts with the very limited display of these behaviours in any pigs undergoing PRT. Additionally, the increasing positive behaviours and low salivary corticosterone levels of the trained pigs indicate this PRT regimen is a practical and welfare-positive tool for the collection of minimally invasive samples from laboratory pigs. While a small-time investment is required to implement this positive-reinforcement training regimen (between 2 and 5 min/pig/day), this is arguably warranted in place of the physical restraint or anaesthesia of pigs required for repeated collection of swabs and rectal temperatures.

## 5. Conclusions

The three-phase positive-reinforcement training (PRT) regimen presented in this study provides a novel option for laboratory pig training for the conscious collection of minimally invasive samples voluntarily. This regimen is most effective in group-housed pigs, where pigs can be trained in six days. Training can be implemented from the day after arrival, allowing for pigs to be trained during the pre-study acclimation period, and no specialist restraint equipment is required as pigs can be trained to have samples collected free standing.

Further refinement of this PRT regimen is recommended to overcome the issue of some pigs being unwilling to enter the Panepinto sling, as observed for one of the seven pigs in the present study. A suggestion for future utilisation is to conduct all training with pigs within the pen rather than training for entry into a Panepinto sling. Pigs could still undergo the three phases of training but standing still within their pen while samples are collected.

## Figures and Tables

**Figure 1 animals-15-00471-f001:**
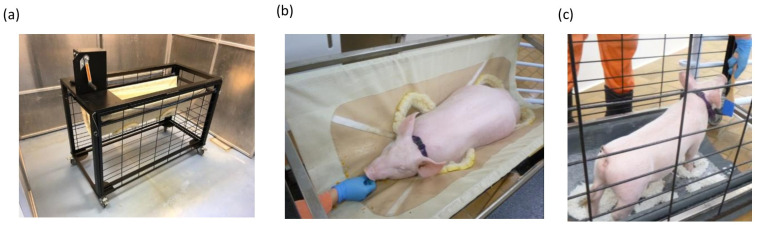
Methods of facilitating voluntary sample collection in laboratory pigs. Image of Panepinto crank-up sling with boat winch mechanism for restraint (**a**), with samples collected from pigs raised in a sling (**b**) or standing still (**c**).

**Figure 2 animals-15-00471-f002:**
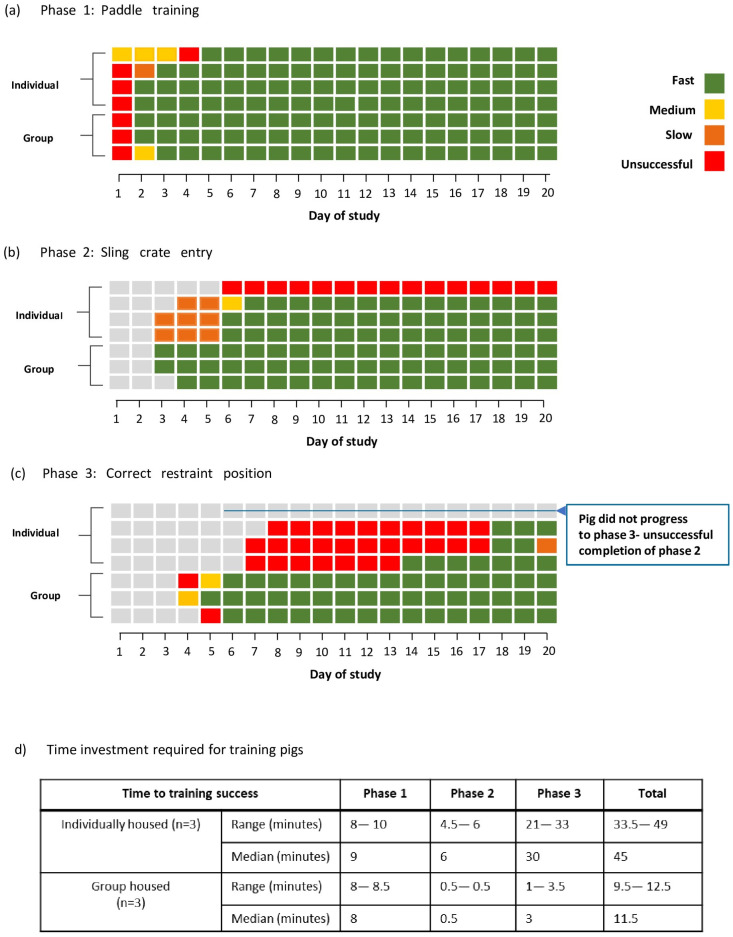
Grouped housing appears to positively impact pig training. Outcomes of pig training are presented schematically, with each line representing an individual pig from either the group-housed or individually housed groups. Coloured boxes represent time taken to perform the trained task and categorised as either fast (0–30 s, green), medium (31–60 s, yellow), slowly (61–180 s, orange) or unsuccessful (>180 s, red) for Phase 1—time to approach (**a**), Phase 2—time to enter pig sling frame (**b**) and Phase 3—time to stand in correct sling position (**c**). Grey boxes represent data from the previous training phase. Time to successfully complete each phase of training was recorded and presented to the nearest 30 s (**d**) for all pigs that successfully completed all three training phases.

**Figure 3 animals-15-00471-f003:**
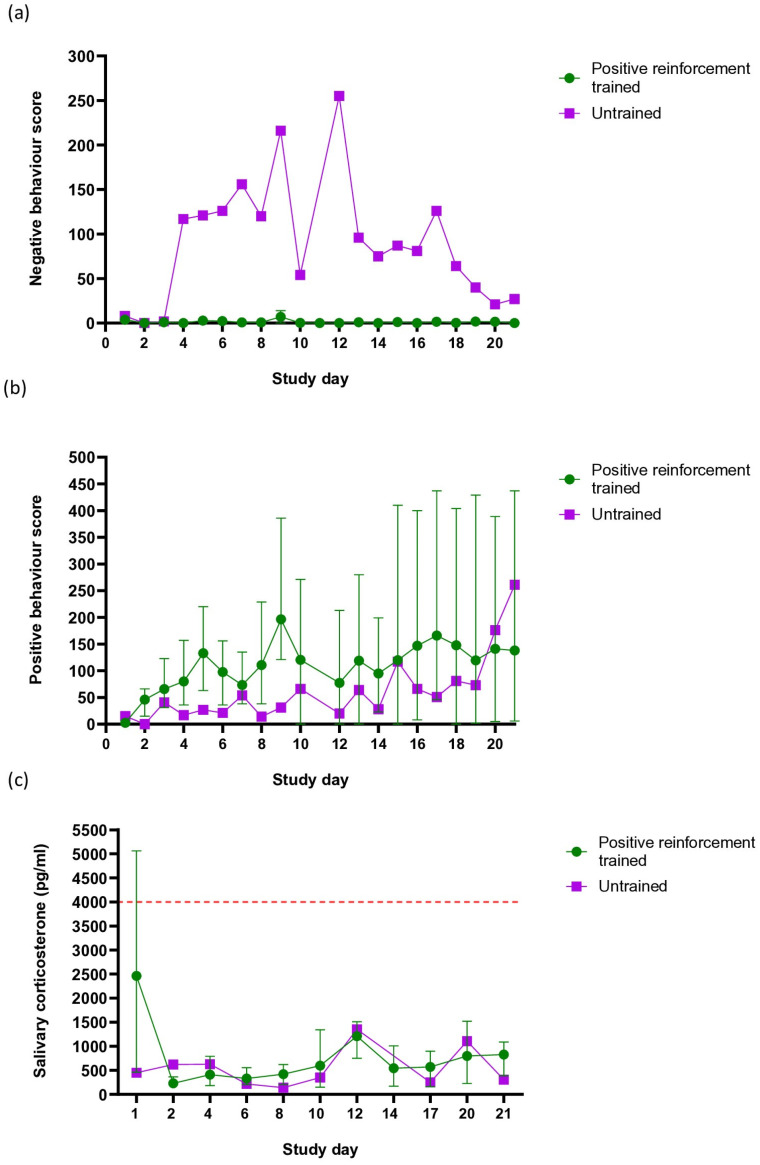
Positive welfare implications of a positive-reinforcement training regime for laboratory pigs. Results of pig training on welfare is presented as negative behaviour score (**a**), positive behaviour score (**b**) and salivary corticosterone (**c**) for trained (n = 4) and untrained (n = 1) individually housed pigs. Red dotted horizontal line indicates maximum level of salivary corticosterone in non-stressed pigs as reported in the published literature. Error bars indicate the range.

**Table 1 animals-15-00471-t001:** A weighted behavioural ethogram for daily assessment of pig behaviour during a human approach test.

Categorisation and Weighting of Behaviour	Behaviour
Low grade positive behaviour—1 point	Sniffing operator—any stretching of the neck towards the operator without making contact Nosing operator—any nose contact to operator Nibbling operator—any gentle open mouth contact with the operator
Medium grade positive behaviour—2 points	Rubbing against operator—any contact with back-and-forth motion on operator Pleasure grunt—any low-pitched verbalisation
High grade positive behaviour—3 points	Play behaviour—any sudden spin or brief fast run away from operator followed by immediate return Tail wagging—any instance of independent circular or side to side tail movement
Low grade negative behaviour—1 point	Bar biting—any instance of biting pen bars
Medium grade negative behaviour—2 points	Fearful retreat—Any sudden movement away from the operator with no return within 10 seconds Fear squeal—Any piercing high-pitched squeal
High grade negative behaviour—3 points	Hard bites any instance of wide mouthed hard biting to operator Open mouthed breathing—pig breathing with mouth open Muscle trembling—any instance of muscle quivering or tremors

## Data Availability

The original contributions presented in this study are included in the article. Further inquiries can be directed to the corresponding author.

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
