# Peer review of "A Positive-Reinforcement Training Regimen for Refined Sample Collection in Laboratory Pigs"

_animals, 2025, doi:10.3390/ani15040471_

Round 1
Reviewer 1 Report
Comments and Suggestions for Authors
This paper presents a short training protocol to train pigs for sample collection using positive reinforcement training. This is an important refinement that will benefit research pigs. I have a few clarifying questions and comments for the authors.
Abstract:
L34: “behavioral ethogram” is unclear here. Was the ethogram used to assess the human approach test or some other behavior at a different time point? Please clarify in the abstract.
L34: suggested changed to “stress response was assessed by analyzing salivary corticosterone” or similar.
L36: What type of sample collection?
Introduction:
Manual restraint is also common for minipigs, which poses injury risk to human handlers and the pigs. It might be worth talking about that briefly as well.
Materials and methods:
Were the pigs housed on floor pens or raised pens? Did they have to step down out of a pen or use a ramp, or was the pen level with the floor?
L90: is the n=3 referring to the number of pens, and if so, was it 18 pigs total? Or number of pigs observed/assessed (one per pen)?
L94: Why just on individually housed pigs?
L102: What was different for the group housed pigs?
L120-121: What was the operator’s behavioral training? Was the operator blinded to pig ID? Was the order of videos observed randomized? Was the operator part of the team conducting the HAT or PRT protocols? Was any intra-observer reliability performed?
L139: Were the cue and food reward initially paired during these training sessions or was this conditioning done prior to the start of the training. Was there a formal pairing of the cue and reward (“charging” of the reinforcer), or did pigs gradually learn this connection during the training? Did pigs have previous experience with the cue and reward prior to this study?
L146: Could the individually housed pigs see each other during the training?
L154: This would be a form of habituation or human socialization, rather than “acclimation”
L160: Along with the marker cue (“yay”)?
L168: Was the pig allowed to touch the paddle while being lured toward the sling? If so, was the same marker and reward given each time the pig touched it? Same question for phase 3 of guiding the pig into the correct position.
L193: How many pigs required that the sling be wound down for the sample, and were the samples still successfully taken? If so, training pigs to stand in a sling to take samples could also be a refinement to actually lifting them up (but harder on the staff!).
Results
Figure 2: Might be helpful to add the criteria for fast, medium, and slow into the figure caption.
L215: Please explain further the unrestrained pigs. At what point in the training was it decided to try this? Was it based on pigs responding poorly to being raised in the sling? How many pigs reacted poorly to being lifted in the sling? Please elaborate on this as I think it is an important refinement.
L219-220: More appropriate to say that “behaviors during a HAT were observed for a single untrained pig…”
L226: “in addition to observations during daily HAT”. An ethogram is a list of behaviors and definitions for recording behavioral observations. You used the ethogram to observe behaviors during the human approach test, it is not an additional test/observation.
Figure 3: What is the Y-axis for the behavior score figures? How were these data analyzed and summarized?
Discussion:
As social creatures, pigs can also learn through social facilitation, in that just observing other pigs completing these tasks and getting rewarded can teach them what to do and that the humans are safe. Could the individually and group housed pigs see each other during the training and HAT?
L255: Habituation or training periods, not acclimation
L270: VAPs could be useful, but testing this PRT regime on voluntarily standing for blood collection could be important too. There are research groups working on this (unpublished), but I’m not sure their training protocol, but it is possible.
L273: The results do not discuss how many pigs were successfully sampled while standing in the sling vs. being lifted. I think these data should be discussed more as that is an important refinement (voluntary sampling).
Conclusions:
L302: The data on free-standing pigs needs to be more clearly presented and discussed if it will be presented in the conclusion.
Author Response
Comment 1: This paper presents a short training protocol to train pigs for sample collection using positive reinforcement training. This is an important refinement that will benefit research pigs. I have a few clarifying questions and comments for the authors.
Response 1: Thank you to Reviewer 1 for their kind and constructive comments, they have improved the manuscript. We are very grateful to them for lending their time and expertise to conduct this review.
Abstract:
Comment 2: L34: “behavioral ethogram” is unclear here. Was the ethogram used to assess the human approach test or some other behavior at a different time point? Please clarify in the abstract.
Response 2: Thank you, this has been changed to ‘Behaviour of individually housed pigs was assessed via an ethogram of behaviours exhibited during a human approach test’ at line 34
Comment 3: L34: suggested changed to “stress response was assessed by analyzing salivary corticosterone” or similar.
Response 3: Thank you, this has been changed at line 34 to ‘and stress response was assessed by analysing salivary corticosterone’
Comment 4: L36: What type of sample collection?
Response 4: Thank you, this has been changed at line 37 to ‘The rapid positive reinforcement training regimen successfully facilitated oral swabbing, rectal swabbing and rectal thermometer insertion’
Introduction:
Comment 5: Manual restraint is also common for minipigs, which poses injury risk to human handlers and the pigs. It might be worth talking about that briefly as well.
Response 5: Thank you, ‘mini-pigs’ has now been specified at line 64, to clarify that issues described for pigs also includes minipigs
Materials and methods:
Comment 6: Were the pigs housed on floor pens or raised pens? Did they have to step down out of a pen or use a ramp, or was the pen level with the floor?
Response 6: Thank you, ‘floor pen’ has been specified at line 101. Additionally, the following has been added to line 183: ‘by stepping over a low horizontal bar (7cm) at the entrance to the sling’
Comment 7: L90: is the n=3 referring to the number of pens, and if so, was it 18 pigs total? Or number of pigs observed/assessed (one per pen)?
Response 7: Thank you, n=3 refers to the number of pigs, housed together in a pen as described: ‘Group housed pigs (n=3) were maintained in a pen measuring 7 x 2.96 m, with a total of six pigs present in the pen.’ These three pigs underwent the positive reinforcement training.
Comment 8: L94: Why just on individually housed pigs?
Response 8: Thank you, the following sentence has been added at line 98 to explain further: ‘Group housed pigs were part of a larger separate study, therefore these variables were not measured’
Comment 9: L102: What was different for the group housed pigs?
Response 9: Thank you, the following has been added to line 112: For group housed pigs, husbandry was conducted (approximately 20 minutes) followed 5 minutes of calm interaction, before commencing 2-5 minutes of PRT per pig.
Comment 10: L120-121: What was the operator’s behavioral training? Was the operator blinded to pig ID? Was the order of videos observed randomized? Was the operator part of the team conducting the HAT or PRT protocols? Was any intra-observer reliability performed?
Response 10: Thank you, ‘an experienced animal technician’ has been added to line 117. The following has also been added to line 129 ‘The operator was not blinded and footage review was not randomised. A second operator was utilised to randomly select HAT footage from one pig per day and conduct the behavioural ethogram, to confirm observer reliability. All assessments were confirmed to be consistent between operators’
Comment 11: L139: Were the cue and food reward initially paired during these training sessions or was this conditioning done prior to the start of the training. Was there a formal pairing of the cue and reward (“charging” of the reinforcer), or did pigs gradually learn this connection during the training? Did pigs have previous experience with the cue and reward prior to this study?
Response 11: Thank you, this has been clarified at line 150 to ‘From phase 1, each time a desired behaviour was performed the pig was positively rewarded…’, which led to rapid association between paddle touching and the reward. The following has been added to line 145 ‘Pigs did not have any prior exposure to positive reinforcement training and could see each other during daily training sessions’
Comment 12: L146: Could the individually housed pigs see each other during the training?
Response 12: Thank you, ‘…and could see each other during daily training sessions’ has been added at line 145.
Comment 13: L154: This would be a form of habituation or human socialization, rather than “acclimation”
Response 13: Thank you, this has been changed to ‘habituation’ at line 168.
Comment 14: L160: Along with the marker cue (“yay”)?
Response 14: Thank, you, this has been added at line 174
Comment 15: L168: Was the pig allowed to touch the paddle while being lured toward the sling? If so, was the same marker and reward given each time the pig touched it? Same question for phase 3 of guiding the pig into the correct position.
Response 15: Thank you, the following has been added for clarification at line 188: ‘A reward was provided once if a pig touched the paddle without stepping forward, after which the pig was only allowed to touch the paddle for a reward if it stepped forward’. Also at line 202 ‘If pigs moved too far back the pig was required to move forward and touch the paddle to receive a reward’
Comment 16: L193: How many pigs required that the sling be wound down for the sample, and were the samples still successfully taken? If so, training pigs to stand in a sling to take samples could also be a refinement to actually lifting them up (but harder on the staff!).
Response 16: Thank you, the following has been added to Results at line 234: ‘Collecting samples from free-standing pigs was more successful than raising pigs in the sling, with only one pig (individually housed) raised daily without struggling’
Results
Comment 17: Figure 2: Might be helpful to add the criteria for fast, medium, and slow into the figure caption.
Response 17: Thank you, the criteria has been added to the figure legend
Comment 18: L215: Please explain further the unrestrained pigs. At what point in the training was it decided to try this? Was it based on pigs responding poorly to being raised in the sling? How many pigs reacted poorly to being lifted in the sling? Please elaborate on this as I think it is an important refinement.
Response 18: Thank you, the following has been added to Results at line 234: ‘Collecting samples from free-standing pigs was more successful than raising pigs in the sling, with only one pig (individually housed) raised daily without struggling. After two failed attempts of raising a pig in the sling, samples were collected free-standing for the remainder of the study’
Comment 19: L219-220: More appropriate to say that “behaviors during a HAT were observed for a single untrained pig…”
Response 19: Thank you, this has been changed to ‘Behaviours during a HAT were also observed for a single untrained pig’ at line 246
Comment 20: L226: “in addition to observations during daily HAT”. An ethogram is a list of behaviors and definitions for recording behavioral observations. You used the ethogram to observe behaviors during the human approach test, it is not an additional test/observation.
Response 20: Thank you, this has been changed to ‘In addition to behavioural observations during the daily HAT via an ethogram….’ At line 254
Comment 21: Figure 3: What is the Y-axis for the behavior score figures? How were these data analyzed and summarized?
Response 21: Thank you, this is described Methods section 2.3: ‘Each behaviour was assigned a score of 1, 2 or 3 based on the weighting described in Table 1, with total daily scores per pig then calculated for both negative and positive behaviours. A behaviour that lasted longer than 5 seconds was counted as a new behaviour to capture its greater significance (eg: longer grunts and squeals)’ Table 1 then details how the behaviour scores were calculated.
Discussion:
Comment 22: As social creatures, pigs can also learn through social facilitation, in that just observing other pigs completing these tasks and getting rewarded can teach them what to do and that the humans are safe. Could the individually and group housed pigs see each other during the training and HAT?
Response 22: Thank you, as suggested at comment 12 we have now added ‘and could see each other during daily training sessions’ at line 145. As all pigs (whether individually or group housed) could observe other pigs during PRT, we believe that this this aspect of social facilitation is unlikely to be a cause of the differences in training times observed between the two housing scenarios.
Comment 23: L255: Habituation or training periods, not acclimation
Response 23: Thank you for raising this. We prefer to use the term ‘acclimation’ here as this is referring to ‘typical pre-study acclimation periods’ that are common for studies involving pigs. Acclimation is the common term utilised in research to describe the period after arrival but prior to the first study procedures commencing, during which period training may or may not be conducted. Therefore, we feel that the use of ‘acclimation’ in this context is preferable.
Comment 24: L270: VAPs could be useful, but testing this PRT regime on voluntarily standing for blood collection could be important too. There are research groups working on this (unpublished), but I’m not sure their training protocol, but it is possible.
Response 24: Thank you, the following has been added at line 288: ‘Further research into potential techniques to facilitate blood collection using this PRT regimen is recommended, such as direct collection from the ear vein for smaller blood volumes and the surgical implantation of vascular access ports for larger blood volumes’
Comment 25: L273: The results do not discuss how many pigs were successfully sampled while standing in the sling vs. being lifted. I think these data should be discussed more as that is an important refinement (voluntary sampling).
Response 25: Thank you for this suggestion, in addition to this aspect being added to Results the following has also been added to Discussion at line 310: ‘By using this training regimen in free-standing pigs, samples can be collected from pigs that choose to voluntarily engage in the sample collection process, demonstrating the welfare benefits of this training regime’
Conclusions:
Comment 26: L302: The data on free-standing pigs needs to be more clearly presented and discussed if it will be presented in the conclusion.
Response 26: Thank you, we feel that this aspect has now been more clearly presented in Results and Discussion.
Reviewer 2 Report
Comments and Suggestions for Authors
This manuscript describes the design and implementation of a simple protocol for positive reinforcement-based training of pigs to be comfortable and cooperative with non-invasive research procedures including oral and rectal swab sample collection and rectal thermometer insertion. The authors are to be congratulated for the effort to improve welfare of research animals. This work nicely highlights the efficiency and safety benefits of employing simple behavioral science learning principles in farm animals.
Introduction: The background, rationale, and general approach are well described.
Methods: My only suggestion (minor) is perhaps a better illustration of the target paddle.
Results: Did you log the total amount of time (minutes) per pig for each phase? I think the range and median number of minutes required for each phase would be informative to those not familiar with how rapidly animals respond to PRT? After all, like all animal welfare improvement challenges, this is primarily a human behavior modification process. My experience has been that protocol decision makers and animal handlers who are convinced of the efficiency and safety advantages are much more successful.
Discussion: Concerning efficiency, can you comment in the discussion on condensing the work into a schedule of fewer days. I expect you could easily get pigs comfortable and cooperating with oral and rectal swabbing and rectal temperature taking on the first day with either/or longer single session or multiple sessions per pig.
Line 306 This paragraph would fit better as the last paragraph of the discussion.
Comments on the Quality of English Language
While most readers will understand your intended meaning, the manuscript can be improved with more precise and appropriate/correct word choice.
“Verbalization” should be “vocalization” (verbalization implied using words)
Throughout the manuscript: Regime” should be “regimen” (regime implies governmental rule, regimen is a plan or protocol)
Throughout the manuscript: Suggest “practical” instead of “feasible.”
Line 69 and elsewhere in manuscript: Suggest “application” or “implementation” or “adoption” rather than “uptake.”
Line 74 and again in conclusions– “to facilitate the collection of minimally-invasive samples consciously and voluntarily.” As written sounds like the pigs were collecting the samples voluntarily and knowingly.
Line 77 – suggest this wording “have oral swabbing rectal swabbing and rectal thermometer insertion”
Line 93 – should be “were performed”
Line 137 – “Every” should be “Each”
Line 138 “Individual” as a noun refers to a person. Suggest “pig” or “animal”
Line 170 Rather than “If a pig refused to move forward…,” “If the pig did not voluntarily move forward…” or "If the pig appeared reluctant to move forward..." (refuse may imply uncooperative)
Line 272 “collected consciously in trained, free-standing pigs” sounds like the operator is conscious. suggest instead “collected in conscious trained, free-standing pigs”
Line 289 “reducing the potential for operator injury” suggest “reducing the risk for operator injury”
Author Response
Comment 1: This manuscript describes the design and implementation of a simple protocol for positive reinforcement-based training of pigs to be comfortable and cooperative with non-invasive research procedures including oral and rectal swab sample collection and rectal thermometer insertion. The authors are to be congratulated for the effort to improve welfare of research animals. This work nicely highlights the efficiency and safety benefits of employing simple behavioral science learning principles in farm animals.
Response 1: Thank you to Reviewer 2 for their kind words and constructive suggestions. Their comments have helped to improve the manuscript, and we are very grateful for their time and expertise.
Comment 2: Introduction: The background, rationale, and general approach are well described.
Response 2: Thank you
Comment 3: Methods: My only suggestion (minor) is perhaps a better illustration of the target paddle.
Response 3: Thank you, we do acknowledge that the image of the paddle, while visible, is small. However, we have chosen to keep the image in figure 1 as it is the best image we have available to demonstrate use of the paddle.
Comment 4: Results: Did you log the total amount of time (minutes) per pig for each phase? I think the range and median number of minutes required for each phase would be informative to those not familiar with how rapidly animals respond to PRT? After all, like all animal welfare improvement challenges, this is primarily a human behavior modification process. My experience has been that protocol decision makers and animal handlers who are convinced of the efficiency and safety advantages are much more successful.
Response 4: Thank you for this valuable suggestion. A table has been added to Figure 2 (d) to capture this information.
Comment 5: Discussion: Concerning efficiency, can you comment in the discussion on condensing the work into a schedule of fewer days. I expect you could easily get pigs comfortable and cooperating with oral and rectal swabbing and rectal temperature taking on the first day with either/or longer single session or multiple sessions per pig.
Response 5: Thank you for this suggestion, we agree and have included at line 288 ‘Future research may be useful to investigate if longer daily training sessions, or multiple training sessions each day, further reduces to time to training success.’
Comment 6: Line 306 This paragraph would fit better as the last paragraph of the discussion.
Response 6: Thank you, this has been moved to the last paragraph of the discussion.
Comments on the Quality of English Language
While most readers will understand your intended meaning, the manuscript can be improved with more precise and appropriate/correct word choice.
Comment 7: “Verbalization” should be “vocalization” (verbalization implied using words)
Response 7: We have checked the manuscript for the use of ‘verbalisation’ and cannot find where this has been used. ‘Verbal cue’ and ‘verbally greeted pigs’ has been used, but this is when describing human use of words for training/ enhancing the human-animal bond.
Comment 8: Throughout the manuscript: Regime” should be “regimen” (regime implies governmental rule, regimen is a plan or protocol)
Response 8: Thank you, ‘regime’ has been changed to ‘regimen’ throughout the manuscript
Comment 9: Throughout the manuscript: Suggest “practical” instead of “feasible.”
Response 9: Thank you, ‘feasible’ has been replaced with ‘practical’ throughout the manuscript
Comment 10: Line 69 and elsewhere in manuscript: Suggest “application” or “implementation” or “adoption” rather than “uptake.”
Response 10: Thank you, ‘uptake’ has been replaced with ‘implementation’ or ‘adoption’ throughout the manuscript.
Comment 11: Line 74 and again in conclusions– “to facilitate the collection of minimally-invasive samples consciously and voluntarily.” As written sounds like the pigs were collecting the samples voluntarily and knowingly.
Response 11: Thank you, the sentence has been changed at line 78 to ‘to allow for minimally-invasive samples to be collected from conscious pigs voluntarily’ and at line 333
Comment 12: Line 77 – suggest this wording “have oral swabbing rectal swabbing and rectal thermometer insertion”
Response 12: Thank you, this has been changed as suggested at lines 37 and 79.
Comment 13: Line 93 – should be “were performed”
Response 13: Thank you, this has been changed at line 97
Comment 14: Line 137 – “Every” should be “Each”
Response 14: Thank you, ‘every’ has been changed to ‘each’ at line 150
Comment 15: Line 138 “Individual” as a noun refers to a person. Suggest “pig” or “animal”
Response 15: Thank you, ‘individual’ has been changed to ‘pig’ at line 151
Comment 16: Line 170 Rather than “If a pig refused to move forward…,” “If the pig did not voluntarily move forward…” or "If the pig appeared reluctant to move forward..." (refuse may imply uncooperative)
Response 16: Thank you, this has been changed to ‘If the pig did not voluntarily move forward’ at line 186
Comment 17: Line 272 “collected consciously in trained, free-standing pigs” sounds like the operator is conscious. suggest instead “collected in conscious trained, free-standing pigs”
Response 17: Thank you, this term has been removed as part of revision of this paragraph at line 306
Comment 18: Line 289 “reducing the potential for operator injury” suggest “reducing the risk for operator injury”
Response 18: Thank you, this has been changed to ‘risk’ at line 325